# FIRAL: An Active Learning Algorithm for Multinomial Logistic Regression

**Youguang Chen**     **George Biros**
Oden Institute for Computational Engineering and Sciences
The University of Texas at Austin

## Abstract

We investigate theory and algorithms for pool-based active learning for multiclass classification using multinomial logistic regression. Using finite sample analysis, we prove that the Fisher Information Ratio (FIR) lower and upper bounds the excess risk. Based on our theoretical analysis, we propose an active learning algorithm that employs regret minimization to minimize the FIR. To verify our derived excess risk bounds, we conduct experiments on synthetic datasets. Furthermore, we compare FIRAL with five other methods and found that our scheme outperforms them: it consistently produces the smallest classification error in the multiclass logistic regression setting, as demonstrated through experiments on MNIST, CIFAR-10, and 50-class ImageNet.

## 1 Introduction

Active learning is of interest in applications with large pools of unlabeled data for which labeling is expensive. In pool active learning, we're given a set of unlabeled points $U$, an initial set of labeled points $S_0$, and a budget of new points $b$, our goal is to algorithmically select $b$ new points to label in order to minimize the log-likelihood error over the unlabeled points. Equivalently instead of selecting points directly, we seek to find a probability density function that we can use to sample the $b$ points. Informally (precise formulation in § 2), let $x$ denote a data point and $p(x)$ denote the distribution density of unlabeled points. Let $q(x)$ be the sampling distribution we will use to select the new $b$ points for labeling. We will choose $q(x)$ in order to minimize the generalization error (or excess risk) of the classifier over $p(x)$. Our theory is classifier specific: it assumes multinomial logistic regression with parameters $\theta$. The expectations of the Hessian—with respect to $\theta$—of the classifier loss function over $p(x)$ and $q(x)$ distributions are denoted by $\mathbf{H}_p$ and $\mathbf{H}_q$ respectively. Using finite sample analysis, our first result (Theorem 3 in § 3) is to show that the unlabeled data excess risk is bounded below and above by the *Fisher information ratio* Trace$(\mathbf{H}_q{}^{-1}\mathbf{H}_p)$, subject to the assumption of $p$ being a sub-Gaussian distribution. Our second result (Theorem 4 in § 4) is to propose and analyze a point selection algorithm based on regret minimization that allows us to bound the generalization error.

There is a large body of work on various active learning methods based on uncertainty estimation ([1, 2, 3]), sample diversity ([4, 5]), Bayesian inference ( [6, 7]), and many others ([8]). Here we just discuss the papers closest to our scheme. Zhang and Oles [9] claimed without proof that FIR is asymptotically proportional to the log-likelihood error of unlabeled data. Sourati et al. [10] proved that FIR is an upper bound of the expected variance of the asymptotic distribution of the log-likelihood error. Chaudhuri et al. [11] proved non-asymptotic results indicating that FIR is closely related to the expected log-likelihood error of an Maximum Likelihood Estimation-based classifier in bounded domain. In this work, we use finite sample analysis to establish FIR-based bounds for the excess risk in the case of multinomial logistic regression with sub-Gaussian assumption for the point distributions.

37th Conference on Neural Information Processing Systems (NeurIPS 2023).

Algorithmically finding points to minimize FIR is an NP-hard combinatorial optimization problem. There have been several approximate algorithms proposed for this problem. Hoi et al. [12] studied the binary classification problem and approximated the FIR using a submodular function and then used a greedy optimization algorithm. Chaudhuri et al. [11] proposed an algorithm that first solves a relaxed continuous convex optimization problem, followed by randomly sampling from the weights. Although they derived a performance guarantee for their approach, it needs a substantial number of samples to approach near-optimal performance solely through random sampling from the weights, and no numerical experiments results were provided using such approach. Ash et al. [13] adopted a forward greedy algorithm to initially select an excess of points and then utilized a backward greedy algorithm to remove surplus points. But such approach has no performance guarantee. Hence, there is still a need for computationally efficient algorithms that can optimize FIR in a multi-class classification context while providing provable guarantees.

Our proposed algorithm, FIRAL, offers a locally near-optimal performance guarantee in terms of selecting points to optimize FIR. In our algorithm we have two steps: first we solve a continuous convex relaxation of the original problem in which we find selection weights for all points in $U$. Then given these weights, we select $b$ points for labeling by a regret minimization approach. This two-step scheme is inspired by Allen-Zhu et al. [14] where a similar approach was used selecting points for linear regression. Extending this approach to active learning for multinomial logistic regression has two main challenges. Firstly, we need to incorporate the information from previously selected points in each new round of active learning. Additionally, while the original approach selects features of individual points, in logistic regression, we need to select a Fisher information matrix ($\mathbf{H}_q$), which complicates the computation and derivation of theoretical performance guarantees. In Section 4, we present our approach in addressing these challenges.

**Our Contributions.** ❶ In § 3 we prove that FIR is a lower and upper bound of the excess risk for multinomial logistic regression under sub-Gaussian assumptions. ❷ In § 4 we detail our FIR Active Learning algorithm (FIRAL) and prove it selects $b$ points that lead to a bound to the excess risk. ❸ In § 5 we evaluate our analysis empirically on synthetic and real world datasets: MNIST, CIFAR-10, and ImageNet using a subset of 50 classes. We compare FIRAL with several other methods for pool-based active learning.

## 2 Problem Formulation

We denote a labeled sample as a pair $(x, y)$, where $x \in \mathbb{R}^d$ is a data point, $y \in \{1, 2, \cdots, c\}$ is its label, and $c$ is the number of classes. Let $\theta \in \mathbb{R}^{(c-1) \times d}$ be the parameters of a $c$-class logistic regression classifier. Given $x$ and $\theta$, the likelihood of the label $y$ is defined by

$$p(y|x, \theta) = \begin{cases} \frac{\exp(\theta_y^\top x)}{1 + \sum_{l \in [c-1]} \exp(\theta_l^\top x)}, & y \in [c-1] \\ \frac{1}{1 + \sum_{l \in [c-1]} \exp(\theta_l^\top x)}, & y = c. \end{cases} \tag{1}$$

We use the negative log-likelihood as the loss function: $\ell_{(x,y)}(\theta) \triangleq -\log p(y|x, \theta)$. To simplify notation we define $\widetilde{d} = d(c-1)$. We derive standard expressions for the gradient $\nabla \ell_{(x,y)}(\theta) \in \mathbb{R}^{(c-1) \times d}$ and Hessian $\nabla^2 \ell_{(x,y)}(\theta) \in \mathbb{R}^{\widetilde{d} \times \widetilde{d}}$ in the Appendix B.1 (Proposition 23). We assume there exists $\theta_*$ such that $p(y|x) = p(y|x, \theta_*)$. Then, given $p(x)$, the joint $(x, y)$ distribution is given by

$$\pi_p(x, y) = p(y|x, \theta_*) p(x). \tag{2}$$

Then, the expected loss at $\theta$ is defined by

$$L_p(\theta) \triangleq \mathbb{E}_{(x,y) \sim \pi_p}[\ell_{(x,y)}(\theta)] = \mathbb{E}_{x \sim p(x)} \mathbb{E}_{y \sim p(y|x, \theta_*)}[\ell_{(x,y)}(\theta)]. \tag{3}$$

The excess risk of $p(x)$ at $\theta$ is defined as $R_p(\theta) = L_p(\theta) - L_p(\theta_*)$. Note that $R_p(\theta) \geq 0$.

**Notation:** The inner product between two matrices is $\mathbf{A} \cdot \mathbf{B} = \text{Trace}(\mathbf{A}^\top \mathbf{B})$. For a matrix $\mathbf{A} \in \mathbb{R}^{m \times n}$, let $\|\mathbf{A}\|$ be the spectral norm of $\mathbf{A}$, let $\text{vec}(\mathbf{A}) \in \mathbb{R}^{mn}$ be the vectorization of $\mathbf{A}$ by stacking all rows together, i.e. $\text{vec}(\mathbf{A}) = (\mathbf{A}_1^\top, \cdots, \mathbf{A}_m^\top)^\top$ where $\mathbf{A}_i$ is $i$-th row of $\mathbf{A}$. Given a positive definite matrix $\mathbf{A} \in \mathbb{R}^{d \times d}$, we define norm $\| \cdot \|_{\mathbf{A}}$ for $x \in \mathbb{R}^d$ by $\|x\|_{\mathbf{A}} = \sqrt{x^\top \mathbf{A} x}$. For

integer $k \geq 1$, we denote by $\mathbf{I}_k$ the $k$-by-$k$ identity matrix. For any point distribution $p(x)$ we define $\mathbf{V}_p \triangleq \mathbb{E}_{x \sim p(x)}[xx^\top]$ to be its covariance matrix, $\mathbf{H}_p(\theta) \triangleq \nabla^2 L_p(\theta)$ be the Hessian matrix of $L_p(\theta)$, define $\mathbf{H}_p \triangleq \mathbf{H}_p(\theta_*)$.

**Active learning.** Let $U = \{x_i\}_{i=1}^m$, be the set of unlabeled points and $S_0$ be the set of $n_0$ initially labeled samples. In particular, we denote the set of points in $S_0$ as $X_0$. Let $\theta_0$ be the solution of training a classifier with $S_0$, i.e., $\theta_0 \in \arg\min_\theta \frac{1}{n_0} \sum_{(x,y) \in S_0} \ell_{(x,y)}(\theta)$. We select a set of $b$ points $X \subset U$, query their labels $y \sim p(y|x, \theta_*), \forall x \in X$, and train a new classifier $\theta_n \in \arg\min_\theta \frac{1}{n} \sum_{(x,y) \in S} \ell_{(x,y)}(\theta)$, where $S$ is the set of $S_0$ with new labeled points and $n = n_0 + b$.

Our goal is to optimize the selection of $X$ so that we can minimize the excess risk on the original unlabeled set $U$, i.e. $L_p(\theta_n) - L_p(\theta_*)$. In this context, we define two problems:

Problem 1: Given $X$ or equivalently $q(x)$, can we bound $L_p(\theta_n) - L_p(\theta_*)$?
Problem 2: Can we construct an efficient algorithm for finding $X$ that minimizes $L_p(\theta_n) - L_p(\theta_*)$?

## 3 Excess Risk Bounds

In this section, we develop our theory to address Problem 1. Our plan is to endow $p(x)$ and $q(x)$ with certain properties (sub-Gaussianity or finite support) and derive FIR bounds for $L_p(\theta_n) - L_p(\theta_*)$. Let $\theta_n$ be the empirical risk minimizer (ERM) obtained from $n$ i.i.d. samples drawn from $\pi_q(x, y)$:

$$\theta_n \in \arg\min_\theta \frac{1}{n} \sum_{i=1}^n \ell_{(x_i, y_i)}(\theta), \qquad \forall i \in [n], \quad (x_i, y_i) \overset{\text{i.i.d.}}{\sim} \pi_q(x, y). \tag{4}$$

We assume that both $p(x)$ and $q(x)$ are sub-Gaussian distributions. Appendix A.1 gives a brief review of definitions and basic properties of sub-Gaussian random variables (vector). We define the $\psi_2$-norm of a sub-Gaussian random variable $x \in \mathbb{R}$ as $\|x\|_{\psi_2} \triangleq \inf\{t > 0 : \mathbb{E}\exp(x^2/t^2) \leq 2\}$. For a sub-Gaussian random vector $x \in \mathbb{R}^d$, $\|x\|_{\psi_2} = \sup\{\|u^\top x\|_{\psi_2} : \|u\|_2 \leq 1\}$. We formalize our assumption for $p(x)$ and $q(x)$ in Assumption 1. Based on this assumption, we can derive some properties for the gradient and Hessian of $\ell_{(x,y)}(\theta)$ shown in Lemma 2 (proof can be found in Appendix D). We present the results for $q$ (thus the subscript in the $K$ constants); exactly the same results, with different constants hold for $p$.

**Assumption 1.** *Let $q(x)$ be a sub-Gaussian distribution for $x \in \mathbb{R}^b$, we assume that $\mathbf{V}_q$ is positive definite. We assume that there exists $r \gtrsim 1$ such that for any $\theta \in \mathcal{B}_r(\theta_*) = \{\theta : \|\theta - \theta_*\|_{2,\infty} \leq r\}$, $\mathbf{H}_q(\theta)$ is positive definite, where $\|\cdot\|_{2,\infty}$ denotes the maximum row norm of a matrix.*

**Lemma 2.** *If Assumption 1 holds for $q(x)$, then for $(x, y) \sim \pi_q(x, y)$:*

*(1) There exists $K_{0,q} > 0$ s.t. $\|\mathbf{V}_q^{-1/2} x\|_{\psi_2} \leq K_{0,q}$.*
*(2) There exists $K_{1,q} > 0$ s.t. $\|\mathbf{H}_q^{-1/2} \text{vec}(\nabla \ell_{(x,y)}(\theta_*))\|_{\psi_2} \leq K_{1,q}$.*
*(3) There exists $K_{2,q}(r) > 0$ s.t. for any $\theta$ in the ball $\mathcal{B}_r(\theta_*) = \{\theta : \|\theta - \theta_*\|_{2,\infty} \leq r\}$,*

$$\sup_{u \in \mathcal{S}^{\widetilde{d}-1}} \|u^\top \mathbf{H}_q(\theta)^{-1/2} \nabla^2 \ell_{(x,y)}(\theta) \mathbf{H}_q(\theta)^{-1/2} u\|_{\psi_1} \leq K_{2,q}(r), \tag{5}$$

*where $\mathcal{S}^{\widetilde{d}-1}$ is the unit sphere in $\mathbb{R}^{\widetilde{d}}$, norm $\|\cdot\|_{\psi_1}$ is the norm defined for a sub-exponential random variable $z \in \mathbb{R}$ by $\|z\|_{\psi_1} = \inf\{t > 0 : \mathbb{E}\exp(|z|/t) \leq 2\}$.*

Our main result of this section is Theorem 3. Under the sample bound given by Eq. (6), we derive high probability bounds for the excess risk in Eq. (7). Details and the proof of Theorem 3 can be found in Appendix C.

**Theorem 3.** *Suppose Assumption 1 holds for both $p(x)$ and $q(x)$. Let $\sigma$ and $\rho > 0$ be constants such that $\mathbf{H}_p \preceq \sigma \mathbf{H}_q$ and $\mathbf{I}_{c-1} \otimes \mathbf{V}_p \preceq \rho \mathbf{H}_p(\theta_*)$ hold. There exit constants $C_1, C_2$ and $C_3 > 0$, such that for any $\delta \in (0, 1)$, whenever*

$$n \geq \max\left\{C_1 \widetilde{d} \log(ed/\delta), \, C_2 \sigma \rho \left(\widetilde{d} + \sqrt{\widetilde{d}} \log(e/\delta)\right)\right\}, \tag{6}$$

where $\widetilde{d} \triangleq d(c-1)$, we have with probability at least $1 - \delta$,

$$\frac{e^{-\alpha} + \alpha - 1}{\alpha^2} \frac{\mathbf{H}_q^{-1} \cdot \mathbf{H}_p}{n} \lesssim \mathbb{E}[L_p(\theta_n)] - L_p \lesssim \frac{e^{\alpha} - \alpha - 1}{\alpha^2} \frac{\mathbf{H}_q^{-1} \cdot \mathbf{H}_p}{n}. \tag{7}$$

Here $\mathbf{H}_p = \mathbf{H}_p(\theta_*)$ and $\mathbf{H}_q = \mathbf{H}_q(\theta_*)$; and $\mathbb{E}$ is the expectation over $\{y_i \sim p(y_i|x_i, \theta_*)\}_{i=1}^n$. Furthermore,

$$\alpha = C_3 \sqrt{\sigma \rho} \sqrt{\left(\widetilde{d} + \sqrt{\widetilde{d}} \log(e/\delta)\right)/n}. \tag{8}$$

From Eq. (7), we observe that FIR ($\mathbf{H}_q^{-1} \cdot \mathbf{H}_p$) appears in both the lower and upper bounds for $R(\theta_n)$. In other words, it is essential for controlling the excess risk. To the right we show how the prefactors $\frac{e^{\alpha} - \alpha - 1}{\alpha^2}$ and $\frac{e^{-\alpha} + \alpha - 1}{\alpha^2}$ change as a function of $\alpha$. Constants $C_1, C_2$ and $C_3$ depend on constants defined in Lemma 2 for both $p(x)$ and $q(x)$. In Appendix D, we derive bounds for $K_{1,p}$ and $K_{2,p}(r)$ in Proposition 35. For a Gaussian design $x \sim \mathcal{N}(0, \mathbf{V}_p)$, we derive bounds for $\rho, K_{0,p}, K_{1,p}$ and $K_{2,p}(r)$ in Proposition 37.

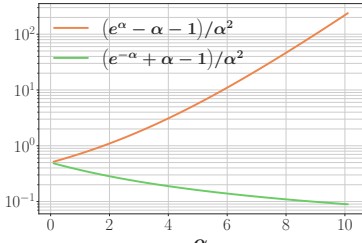

Figure 1: *FIR prefactors in Eq. (7).*

**Bounded domain.** If the domain of $x$ is bounded, Chaudhuri et al.[11] provided lower and upper bounds for the excess risk of $p(x)$ (Lemma 1 in [11]). Their conclusion is similar to ours, namely that FIR is crucial in controlling the excess risk of $p(x)$. It is worth noting that when the domain is bounded, both $p(x)$ and $q(x)$ are always sub-Gaussian. Thus, our assumption is more general. For the sake of completeness, we provide a detailed derivation of the excess risk bounds for $p(x)$ in Theorem 40 when $x$ is bounded with Assumption 38.

## 4 Active Learning via Minimizing Fisher Information Ratio

We now discuss the FIRAL algorithm that addresses Problem 2. We can use the theoretical analysis derived in the previous section to guide us for the point selection. Let $p(x)$ be the empirical distribution on unlabeled pool $U$ with $|U| = m$, and $q$ the distribution for the $n = n_0 + b$ labeled points. Eq. (7) inspires us to select points to label such that we can minimize the FIR $\mathbf{H}_q^{-1} \cdot \mathbf{H}_p$, where $\mathbf{H}_q = \mathbf{H}_q(\theta_*), \mathbf{H}_p = \mathbf{H}_p(\theta_*)$. However, we cannot directly use this as $\theta_*$ is unknown. Instead, we will use $\theta_0$, the solution by training the classifier with the initial labeled set $S_0$.[1] That is, we will find $q$ by minimizing $\mathbf{H}_q(\theta_0)^{-1} \cdot \mathbf{H}_p(\theta_0)$.

In § 4.1, we formalize our optimization objective in Eq. (13). Solving Eq. (13) exactly is NP-hard [15]. Inspired by [14], we approximate the solution in two steps: *(1) we solve a continuous convex optimization problem in Eq. (14) (§ 4.2), (2) and use the results in a regret minimization algorithm to select points by Eq. (19) (§ 4.3)*. In Algorithm 1 we summarize the scheme.

We state theoretical guarantees for the algorithm in § 4.4, where we prove that it achieves $(1 + \epsilon)$-approximation of the optimal objective value in Eq.(13) with sample complexity $b = \mathcal{O}(\tilde{d}/\epsilon^2)$, as stated in Theorem 10. Finally, we obtain the excess risk bound for unlabeled points $U$ by accounting for the fact that we use $\theta_0$ instead of $\theta_*$ in the objective function. The overall result is summarized in the following theorem.

**Theorem 4.** *Suppose that Assumption 1 holds. Let $\epsilon \in (0, 1)$, $\delta \in (0, 1)$, and b the number of points to label. Then with probability at least $1 - \delta$, the $\theta_n$—computed by fitting a multinomial logistic regression classifier on the labeled points selected using FIRAL (Algorithm 1) with learning rate $\eta = 8\sqrt{\widetilde{d}}/\epsilon$, $b \geq 32\widetilde{d}/\epsilon^2 + 16\sqrt{\widetilde{d}}/\epsilon^2$, and $n = n_0 + b$ satisfying Eq. (6)—results in*

$$\mathbb{E}[L_p(\theta_n)] - L_p(\theta_*) \lesssim (1 + \epsilon) \, 2e^{2\alpha_0} \frac{e^{\alpha_n} - \alpha_n - 1}{\alpha_n^2} \frac{OPT}{n}. \tag{9}$$

*Here OPT is the minimal $\mathbf{H}_q^{-1} \cdot \mathbf{H}_p$, attained by selecting the best b points from $U$; $\mathbb{E}$ is the expectation over $\{y_i \sim p(y_i|x_i, \theta_*)\}_{i=1}^n$; and $\alpha_0$ and $\alpha_n$ are constants.*

---

[1] Such a set can be generated by an alternative method that only uses $p(x)$ to select points, e.g., K-means.

### 4.1 Optimization objective

First we define the precise expression for $\mathbf{H}_q(\theta_0)^{-1} \cdot \mathbf{H}_p(\theta_0)$. We define the Fisher information matrix $\mathbf{H}(x, \theta) = \nabla^2 \ell_{(x,y)}(\theta)$. By Eq. (40) in Proposition 23 (Appendix B), we find that for multinomial logistic regression

$$\mathbf{H}(x, \theta) = \left[\text{diag}(\mathbf{h}(x, \theta)) - \mathbf{h}(x, \theta)\mathbf{h}(x, \theta)^\top\right] \otimes (xx^\top), \tag{10}$$

where $\otimes$ represents the matrix Kronecker product, $\mathbf{h}(x, \theta)$ is a $(c-1)$-dimensional vector whose $k$-th component is $\mathbf{h}_k(x, \theta) = p(y = k | x, \theta)$. In Eq. (10) we can see that the Hessian of $\ell_{(x,y)}(\theta)$ does not depend on the class label $y$. Following our previous definitions, $\mathbf{H}_p(\theta_0) = \nabla^2 L_p(\theta_0) = \frac{1}{m}\sum_{x \in U}\mathbf{H}(x, \theta_0)$ and $\mathbf{H}_q(\theta_0) = \nabla^2 L_q(\theta_0) = \frac{1}{n}\sum_{x \in X_0 \cup X}\mathbf{H}(x, \theta_0)$. For notational simplicity we also define

$$\mathbf{H}(x) \triangleq \mathbf{H}(x, \theta_0) + \frac{1}{b}\sum_{x' \in X_0}\mathbf{H}(x', \theta_0) \tag{11}$$

$$\boldsymbol{\Sigma}(z) \triangleq \sum_{i \in [m]} z_i \mathbf{H}(x_i), \quad z_i \text{ scalar.} \tag{12}$$

Then minimizing $\mathbf{H}_q(\theta_0)^{-1} \cdot \mathbf{H}_p(\theta_0)$ is equivalent to

$$\min_{\substack{z \in \{0,1\}^m \\ \|z\|_1 = b}} f(z) \triangleq f\left(\boldsymbol{\Sigma}(z)\right) \triangleq \left(\boldsymbol{\Sigma}(z)\right)^{-1} \cdot \mathbf{H}_p(\theta_0). \tag{13}$$

We define $z_*$ be the optimal solution of Eq. (13) and $f_* \triangleq f(z_*)$. In the following, with some abuse of notation, we will consider $f$ being a function of either a vector $z$ or a positive semidefinite matrix $f(\boldsymbol{\Sigma})$, depending on the context. Lemma 5 lists key properties of $f$ when viewed as a matrix function; we use them in § 4.2 to prove the optimality of FIRAL.

**Lemma 5.** $f : \{\mathbf{A} \in \mathbb{R}^{\tilde{d} \times \tilde{d}} : \mathbf{A} \succeq \mathbf{0}\} \to \mathbb{R}$ defined in Eq. (13) satisfies the following properties:

(1) convex: $f(\lambda\mathbf{A} + (1-\lambda)\mathbf{B}) \leq \lambda f(\mathbf{A}) + (1-\lambda)f(\mathbf{B})$ for all $\lambda \in [0,1]$ and $\mathbf{A}, \mathbf{B} \in \mathbb{S}_{++}^{\tilde{d}}$
(2) monotonically non-increasing: if $\mathbf{A} \preceq \mathbf{B}$ then $f(\mathbf{A}) \geq f(\mathbf{B})$,
(3) reciprocally linear: if $t > 0$ then $f(t\mathbf{A})) = t^{-1}f(\mathbf{A})$.

### 4.2 Relaxed problem

As a first step in solving Eq. (13) we relax the constraint $z \in \{0,1\}^m$ to $z \in [0,1]^m$. Then we obtain the following convex programming problem:

$$z_\diamond = \arg\min_{\substack{z \in [0,1]^m \\ \|z\|_1 = b}} f(\boldsymbol{\Sigma}(z)). \tag{14}$$

Since both the objective function and the constraint set are convex, conventional convex programming algorithm can be used to solve Eq. (14). We choose to use a mirror descent algorithm in our implementation (outlined in the Appendix, Algorithm 2). Since the integrality constraint is a subset of the relaxed constraint we obtain the following result.

**Proposition 6.** $f(z_\diamond) \leq f_*$.

In what follows, we use matrices $\boldsymbol{\Sigma}_\diamond$ and $\widetilde{\mathbf{H}}(x_i)$ ($i \in [m]$) defined by

$$\boldsymbol{\Sigma}_\diamond \triangleq \boldsymbol{\Sigma}(z_\diamond) \quad \text{and} \quad \widetilde{\mathbf{H}}(x_i) \triangleq \boldsymbol{\Sigma}_\diamond^{-1/2}\mathbf{H}(x_i)\boldsymbol{\Sigma}_\diamond^{-1/2}, \quad i \in [m]. \tag{15}$$

### 4.3 Solving Sparsification problem via Regret Minimization

**Goal of sparsification.** Now we introduce our method of sparsifying $z_\diamond$ (optimal solution to Eq. (14)) into a valid integer solution to Eq. (13). To do so, we use an online optimization algorithm in which we select one point at a time in sequence until we have $b$ points. Notice that alternative techniques like thresholding $z_\diamond$ could be used but it was unclear to us how to provide error estimates for such a scheme. Instead, we use an alternative scheme that we describe below.

Let $i_t \in [m]$ be the point index selected at step $t \in [b]$. We can observe that if $\lambda_{\min}\left(\sum_{t\in[b]} \widetilde{\mathbf{H}}(x_{i_t})\right) \geq \tau$ for some $\tau > 0$, then $\sum_{t\in[b]} \mathbf{H}(x_{i_t}) \succeq \tau\boldsymbol{\Sigma}_\diamond$. By Lemma 5-Item (3) and Proposition 6, we obtain the following result.

**Proposition 7.** *Given $\tau \in (0,1)$, we have*

$$\lambda_{\min}\left(\sum_{t\in[b]} \widetilde{\mathbf{H}}(x_{i_t})\right) \geq \tau \Longrightarrow f\left(\sum_{t\in[b]} \mathbf{H}(x_{i_t})\right) \leq \tau^{-1} f_*. \tag{16}$$

From Eq. (16), a larger $\tau$ value indicates that $f$ is closer to $f_*$. Therefore, our objective is to choose points in such a way that $\lambda_{\min}\left(\sum_{t\in[b]} \widetilde{\mathbf{H}}(x_{i_t})\right)$ is maximized.

**Lower bound minimum eigenvalue via Follow-The-Regularized-Leader (FTRL).** We apply FTRL, which is a popular method for online optimization [16], to our problem because it can yield a lower bound for $\lambda_{\min}\left(\sum_{t\in[b]} \widetilde{\mathbf{H}}(x_{i_t})\right)$ in our setting. FTRL takes $b$ steps to finish. At each step $t \in [b]$, for a fixed learning rate $\eta > 0$, we generate a matrix $\mathbf{A}_t$ defined by

$$\mathbf{A}_1 = \frac{1}{\widetilde{d}}\mathbf{I}_{\widetilde{d}}, \qquad \mathbf{A}_t = \left(\nu_t \mathbf{I}_{\widetilde{d}} + \eta \sum_{l=1}^{t-1} \widetilde{\mathbf{H}}(x_{i_l})\right)^{-2} \quad (t \geq 2). \tag{17}$$

Here $\nu_t$ is the unique constant such that $\mathrm{Trace}(\mathbf{A}_t) = 1$. Using Eq. (17) we can guarantee a lower bound for $\lambda_{\min}\left(\sum_{s\in[t]} \widetilde{\mathbf{H}}(x_{i_t})\right)$, which is formalized below:

**Proposition 8.** *Given $A_l$, $l \in [b]$, defined by Eq. (17) and for all $t \in [b]$*

$$\lambda_{\min}\left(\sum_{l=1}^t \widetilde{\mathbf{H}}(x_{i_l})\right) \geq -\frac{2\sqrt{\widetilde{d}}}{\eta} + \frac{1}{\eta}\sum_{l=1}^t \mathrm{Trace}\left[\mathbf{A}_l^{1/2} - \left(\mathbf{A}_l^{-1/2} + \eta\widetilde{\mathbf{H}}(x_{i_l})\right)^{-1}\right]. \tag{18}$$

**Point selection via maximizing the lower bound in Eq. (18).** Now we discuss our choice of point selection at each time step based on Eq. (18). Recall that our sparsification goal is to make $\lambda_{\min}(\sum_{s=1}^t \widetilde{\mathbf{H}}(x_{i_s})$ as large as possible. Since Eq. (18) provides a lower bound for such minimum eigenvalue, we can choose $i_t \in [m]$ to maximize the lower bound, which is equivalent to choose

$$i_t \in \underset{i\in[m]}{\arg\min}\, \mathrm{Trace}\left[\left(\mathbf{A}_t^{-1/2} + \eta\widetilde{\mathbf{H}}(x_i)\right)^{-1}\right]. \tag{19}$$

Solving Eq. (19) directly can become computationally expensive when the dimension $d$, number of classes $c$, and the pool size $n$ are large. This is due to the fact that the matrix $\mathbf{A}_t^{-1/2} + \eta\widetilde{\mathbf{H}}(x_i) \in \mathbb{R}^{\widetilde{d}\times\widetilde{d}}$ (where $\widetilde{d} = d(c-1)$), requiring $n$ eigendecompositions of a $\widetilde{d} \times \widetilde{d}$ matrix to obtain the solution. Fortunately, we can reduce this complexity *without losing accuracy*. First, by Eq. (10) and Eq. (11), we have for any $i \in [m]$,

$$\mathbf{H}(x_i) = \underbrace{\frac{1}{b}\sum_{x\in X_0}\mathbf{H}(x,\theta_0)}_{\triangleq \mathbf{D}} + \underbrace{\left[\mathrm{diag}(\mathbf{h}(x_i,\theta_0)) - \mathbf{h}(x_i,\theta_0)\mathbf{h}(x_i,\theta_0)^\top\right]}_{\triangleq \mathbf{V}_i\boldsymbol{\Lambda}_i\mathbf{V}_i^\top} \otimes (x_i x_i^\top), \tag{20}$$

where $\mathbf{V}_i\boldsymbol{\Lambda}_i\mathbf{V}_i^\top$ is the eigendecomposition of $\mathrm{diag}(\mathbf{h}(x_i,\theta_0)) - \mathbf{h}(x_i,\theta_0)\mathbf{h}(x_i,\theta_0)^\top$. Define matrix $\mathbf{Q}_i \triangleq \mathbf{V}_i\boldsymbol{\Lambda}_i^{1/2}$, then $\widetilde{\mathbf{H}}(x_i) = \mathbf{D} + (\mathbf{Q}_i\mathbf{Q}_i^\top) \otimes (x_i x_i^\top)$. Substitute this into Eq. (15), we have a new expression for transformed Fisher information matrix $\widetilde{\mathbf{H}}(x_i)$:

$$\widetilde{\mathbf{H}}(x_i) = \underbrace{(\boldsymbol{\Sigma}_\diamond)^{-1/2}\mathbf{D}(\boldsymbol{\Sigma}_\diamond)^{-1/2}}_{\triangleq \widetilde{\mathbf{D}}} + \underbrace{(\boldsymbol{\Sigma}_\diamond)^{-1/2}(\mathbf{Q}_i \otimes x_i)}_{\triangleq \widetilde{\mathbf{P}}_i}(\mathbf{Q}_i \otimes x_i)^\top(\boldsymbol{\Sigma}_\diamond)^{-1/2} = \widetilde{\mathbf{D}} + \widetilde{\mathbf{P}}_i\widetilde{\mathbf{P}}_i^\top. \tag{21}$$

Now define $\mathbf{B}_t \in \mathbb{R}^{\widetilde{d}\times\widetilde{d}}$ s.t. $\mathbf{B}_t^{-1/2} = \mathbf{A}_t^{-1/2} + \eta\widetilde{\mathbf{D}}$. By Eq. (21), we have $\mathbf{A}_t^{-1/2} + \eta\widetilde{\mathbf{H}}(x_i) = \mathbf{B}_t^{-1/2} + \eta\widetilde{\mathbf{P}}_i\widetilde{\mathbf{P}}_i^\top$. Applying Woodbury's matrix identity, we have

$$\left(\mathbf{A}_t^{-1/2} + \eta\widetilde{\mathbf{H}}(x_i)\right)^{-1} = \mathbf{B}_t^{1/2} - \eta\mathbf{B}_t^{1/2}\widetilde{\mathbf{P}}_i\left(\mathbf{I}_{c-1} + \eta\widetilde{\mathbf{P}}_i^\top\mathbf{B}_t^{1/2}\widetilde{\mathbf{P}}_i\right)^{-1}\widetilde{\mathbf{P}}_i^\top\mathbf{B}_t^{1/2}. \tag{22}$$

---

**Algorithm 1** FIRAL($b$, $U$, $S_0$, $\theta_0$)

---
**Input:** sample budget $b$, unlabeled pool $U = \{x_i\}_{i\in[m]}$, labeled set $S_0$, initial ERM $\theta_0$
**Output:** selected points $X$

1: $X \leftarrow \emptyset$
2: $z_\diamond \leftarrow$ solution of Eq. (14), $\boldsymbol{\Sigma}_\diamond \leftarrow \sum_{i=1}^n z_{*,i}\mathbf{H}(x_i)$         # continuous convex relaxation
3: $\mathbf{V}_i\boldsymbol{\Lambda}_i\mathbf{V}_i^\top \leftarrow$ eigendecomposition of $\mathrm{diag}(\mathbf{h}(x_i, \theta_0)) - \mathbf{h}(x_i, \theta_0)\mathbf{h}(x_i, \theta_0)^\top, \forall i \in [m]$
4: $\widetilde{\mathbf{P}}_i \leftarrow \boldsymbol{\Sigma}_\diamond^{-1/2}\big(x_i \otimes (\mathbf{V}_i\boldsymbol{\Lambda}_i^{1/2})\big), \forall i \in [m]$
5: $\widetilde{\mathbf{D}} \leftarrow$ defined in Eq. (21), $\mathbf{A}_1^{-1/2} \leftarrow \sqrt{\widetilde{d}}\mathbf{I}_{\widetilde{d}}$, $\mathbf{B}_1^{1/2} \leftarrow (\mathbf{A}_1^{-1/2} + \eta\widetilde{\mathbf{D}})^{-1}$
6: **for** $t = 1$ to $b$ **do**
7:      $i_t \leftarrow$ solution of Eq. (23), $X \leftarrow X \cup \{x_{i_t}\}$
8:      $\mathbf{V}\boldsymbol{\Lambda}\mathbf{V}^\top \leftarrow$ eigendecomposition of $\eta\sum_{s=1}^t \widetilde{\mathbf{H}}(x_{i_s}) = \eta\sum_{s=1}^t (\widetilde{\mathbf{D}} + \widetilde{\mathbf{P}}_{i_s}\widetilde{\mathbf{P}}_{i_s}^\top)$
9:      find $\nu_{t+1}$ s.t. $\sum_{j\in[\widetilde{d}]}(\nu_{t+1} + \lambda_j)^{-2} = 1$
10:     $\mathbf{A}_{t+1}^{-1/2} \leftarrow \mathbf{V}(\nu_{t+1}\mathbf{I}_{\widetilde{d}} + \boldsymbol{\Lambda})\mathbf{V}^\top$, $\mathbf{B}_{t+1}^{1/2} \leftarrow (\mathbf{A}_{t+1}^{-1/2} + \eta\widetilde{\mathbf{D}})^{-1}$
11: **end for**

---

Now our point selection objective Eq. (19) is equivalent to

$$i_t \leftarrow \underset{i\in[m]}{\arg\max} \left(\mathbf{I}_{c-1} + \eta\widetilde{\mathbf{P}}_i^\top\mathbf{B}_t^{1/2}\widetilde{\mathbf{P}}_i\right)^{-1} \cdot \widetilde{\mathbf{P}}_i^\top\mathbf{B}_t\widetilde{\mathbf{P}}_i. \tag{23}$$

Since $(\mathbf{I}_{c-1} + \eta\widetilde{\mathbf{P}}_i^\top\mathbf{B}_t^{1/2}\widetilde{\mathbf{P}}_i) \in \mathbb{R}^{(c-1)\times(c-1)}$, solving Eq. (23) is faster than solving Eq. (19). We summarize FIRAL for selecting $b$ points in Algorithm 1.

**Connection to regret minimization.** Our algorithm is derived as the solution of a regret minimization problem in the adversarial linear bandits setting. We give a brief introduction in Appendix F.3. Readers who are interested in this topic can refer to Part VI of [17]. In our case the action matrix is constrained to $\{\mathbf{A} \in \mathbb{R}^{\widetilde{d}\times\widetilde{d}} : \mathbf{A} \succeq 0, \mathrm{Trace}(\mathbf{A}) = 1\}$ and is chosen by Eq. (17); the loss matrix is constrained to the set of the transformed Fisher information matrices $\{\widetilde{\mathbf{H}}(x_i)\}_{i=1}^m$ and is chosen by minimizing Eq. (23).

**Algorithm complexity.** Our algorithm has two steps: convex relaxation (line 2 in Algorithm 1) and sparsification (lines 3–11). Let $T_{\mathrm{eigen}}(\widetilde{d})$ be the complexity of eigendecomposition of a $\widetilde{d}$-dimensional symmetric positive definite matrix. Given an unlabeled point pool $U$ with $m = |U|$, the complexity of solving the convex relaxation problem by mirror descent (Algorithm 2) is $\mathcal{O}\big(m\widetilde{d}^2\log m + T_{\mathrm{eigen}}(\widetilde{d})\log m\big)$, where we assume that the number of iterations is $\mathcal{O}(\log m)$ according to Theorem 42. Given sample budget $b$, the complexity of solving the sparsification problem is $\mathcal{O}\big(T_{\mathrm{eigen}}(\widetilde{d})b + T_{\mathrm{eigen}}(c-1)bm\big)$.

## 4.4 Performance guarantee

We intend to lower bound $\lambda_{\min}\big(\sum_{t\in[b]}\widetilde{\mathbf{H}}(x_{i_t})\big)$ through lower bounding the right hand side of (18). First, since our point selection algorithm selects point $x_i$ at each step to maximize $\mathrm{Trace}[\mathbf{A}_t^{1/2} - (\mathbf{A}_t^{-1/2} + \eta\widetilde{\mathbf{H}}(x_i))^{-1}]$, we establish a lower bound for this term at each step, as demonstrated in Proposition 9.

**Proposition 9.** *At each step $t \in [b]$, we have*

$$\max_{i\in[m]}\frac{1}{\eta}\mathrm{Trace}[\mathbf{A}_t^{1/2} - (\mathbf{A}_t^{-1/2} + \eta\widetilde{\mathbf{H}}(x_i))^{-1}] \geq \frac{1 - \frac{\eta}{2b}}{b + \eta\sqrt{\widetilde{d}}}. \tag{24}$$

The derivation is elaborated in Appendix F.4. We remark that there is a similar lower bound derived for the optimal design setting in [14] (Lemma 3.2), where a rank-1 matrix $\tilde{x}_{i_t}\tilde{x}_{i_t}^\top$ ($i_t \in [m]$ and $\tilde{x}_{i_t} \in \mathbb{R}^d$) is selected at each step. In contrast, in our active learning setting, the selected matrix $\widetilde{\mathbf{H}}(x_{i_t})$ possesses a minimum rank of $c - 1$ and can even be a full-rank matrix, contingent upon

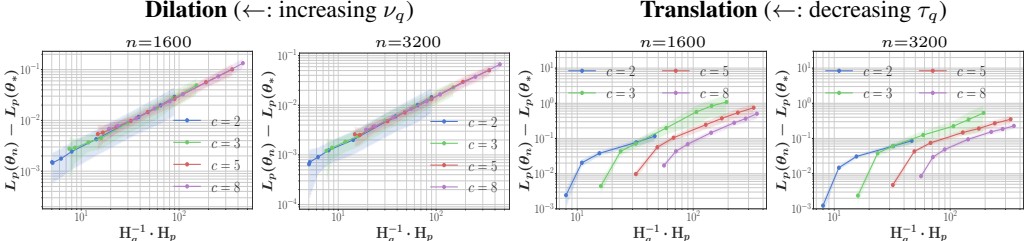

Figure 2: *Synthetic experiments: excess risk of $p(x)$ as a function of the FIR ($\mathbf{H}_q^{-1} \cdot \mathbf{H}_p$) in dilation and translation tests.*

the labeled points from prior rounds. The distinction between the characteristics of the matrices significantly complicates the derivation of such a general lower bound.

By connecting the observations obtained in this section, we can show that our algorithm can achieve $(1 + \epsilon)$-approximation of the optimal objective with sample size $\mathcal{O}(\widetilde{d}/\epsilon^2)$. We conclude our results in Theorem 10.

**Theorem 10.** *Given $\epsilon \in (0,1)$, let $\eta = 8\sqrt{\widetilde{d}}/\epsilon$, whenever $b \geq 32\widetilde{d}/\epsilon^2 + 16\sqrt{\widetilde{d}}/\epsilon^2$, denote the instance index selected by Algorithm 1 at step $t$ by $i_t \in [m]$, then the algorithm is near-optimal: $f\left(\sum_{t \in [b]} \mathbf{H}(x_{i_t})\right) \leq (1 + \epsilon)f_*$, where $f$ is the objective function defined in Eq. (13) and $f_*$ is its optimal value.*

The excess risk upper bound for unlabeled points can be obtained using our algorithm in Theorem 4 by combining Theorem 3 and Theorem 10 while considering the impact of using $\theta_0$ as an approximation for $\theta_*$. We present the proof in Appendix F.6. Comparing Eq.(9) to Eq.(7), we observe a factor of $2(1 + \epsilon)e^{2\alpha_0}$ degradation in the upper bound. The $(1 + \epsilon)$-term comes from our algorithm, while the $2e^{2\alpha_0}$-term comes from the use of $\theta_0$ instead of $\theta_*$. This observation suggests that, given a total budget of points to label $b$ we should consider an iterative approach consisting of $r$ active learning rounds. At each round $k$ we label a new batch of size $b/r$ points and we obtain a new estimate $\theta_k$ that can be used to approximate $\theta_*$. The prefactor containing $\alpha_0$ will becomes $\alpha_k$ and reduces $\theta_k$ converges to $\theta_*$. The simplest solution would be to use $r = b$ but this can be computationally expensive. In our tests, we use this batched approach and choose $b/r$ to be a small multiple of $c$.

## 5 Numerical Experiments

**Synthetic datasets.** We use synthetic datasets to demonstrate the excess risk bounds Eq. (7) derived in Theorem 3. We choose $p(x) \sim \mathcal{N}(\mathbf{0}, \mathbf{V}_p)$, where $\mathbf{V}_p = 100\mathbf{I}_d$ and $d = 8$. We explore different numbers of classes denoted by $c \in \{2, 3, 5, 8\}$. We define the ground truth parameter $\theta_*$ in such a way that the points generated from $p(x)$ are nearly equally distributed across the $c$ classes. In Fig. 4 (Appendix G.1), we plot the first two coordinates of the points draw from $p(x)$, where each point is colored by its class id.

We conduct tests using two different types of $q(x)$ based on operations applied to $p(x)$: *dilation* and *translation*. For the dilation, $q(x) \sim \mathcal{N}(\mathbf{0}, \nu_q \mathbf{V}_p)$, where $\nu_q \in \mathbb{R}^+$. We vary $\nu_q$ within so that FIR ($\mathbf{H}_q^{-1} \cdot \mathbf{H}_p$) is in $[0.2\widetilde{d}, 10\widetilde{d}]$, where $\widetilde{d} = d(c - 1)$. For translation, $q(x) \sim \mathcal{N}(\tau_q \mathbf{a}, \mathbf{V}_p)$, where $\mathbf{a} = (1/\sqrt{2}, 1/\sqrt{2}, 0, \cdots, 0)$ and $\tau_q \in \mathbb{R}^+$. We examine various $\tau_q$ values that ensures $\mathbf{H}_q^{-1} \cdot \mathbf{H}_p \in [\widetilde{d}, 10\widetilde{d}]$. For each $c \in \{2, 3, 5, 8\}$, $q(x)$ and $n \in \{1600, 3200\}$, we i.i.d. draw $n$ samples from $\pi_q(x)$ and obtain $\theta_n$ defined by Eq. (4) using these samples. We estimate excess risk $L_p(\theta_n) - L_p(\theta_*)$ by averaging the log-likelihood error on $5 \times 10^4$ i.i.d. points sampled from $p(x)$.

Fig. 2 displays the excess risk plotted against FIR for both dilation tests (left two plots) and translation tests (right two plots). It is evident that FIR plays a crucial role in controlling the excess risk. In the case of dilation tests, we observe an almost linear convergence rate with respect to FIR. In the translation tests, we observe a faster-than-linear convergence rate, which can be explained by examining the upper bound of Eq. (7). As FIR decreases, $\sigma$ also decreases according to the right plot of Fig. 6). By Proposition 37, in our scenario, we have $K_{1,q} \lesssim (100 + \tau_q)^{3/4}$. In Appendix C,

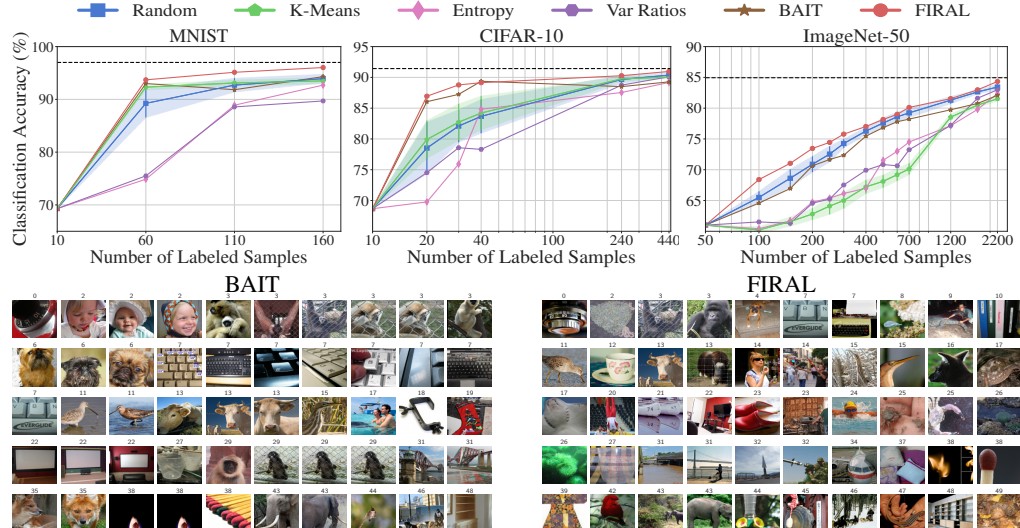

Figure 3: *Active learning results for MNIST (left), CIFAR-10 (center) and ImageNet-50 (right). Black dashed lines in the upper row plots are the classification accuracy using all points in $U$ and their labels. The lower row shows 50 images that are selected in the first round of the active learning process for the ImageNet-50 dataset.*

it is stated that $C_3 = \mathcal{O}(K_{0,p} K_{1,q} K_{2,p})$. As a result, as FIR decreases, both $C_3$ and $\sigma$ decrease, leading to a decrease in $\alpha$ (as indicated by Eq. (63)). Referring to Fig. 1, the prefactor of the FIR term in the upper bound decreases as $\alpha$ decreases. Consequently, the upper bound of Eq. (7) indicates a faster-than-linear convergence rate with respect to the FIR term in the case of translation. We perform similar tests on multivariate Laplace distribution and t-distribution, and the results are consistent with our observations on Gaussian tests. Further details of synthetic experiments are given in Appendix G.1.

**Real-world datasets.** We demonstrate the effectiveness of our active learning algorithm using three real-world datasets: MNIST [18], CIFAR-10 [19], and ImageNet [20]. In the case of ImageNet, we randomly choose 50 classes for our experiments. First we use unsupervised learning to extract features and then apply active learning to the feature space, that is, we do **not** use any label information in our pre-processing. For MNIST, we calculate the normalized Laplacian of the training data and use the spectral subspace of the 20 smallest eigenvalues. For CIFAR-10 and ImageNet-50, we use a contrastive learning SimCLR model [21]; then we compute the normalized nearest-neighbor Laplacian and select the subspace of the 20 smallest eigenvalues; For ImageNet-50 we select the subspace of the 40 smallest eigenvalues. For each dataset, we initialize the labeled data $S_0$ by randomly selecting one sample from each class. Further details about tuning hyperparameter $\eta$ and data pre-processing are given in Appendix G.2.

We compare our algorithm FIRAL with five methods: (1) Random selection, (2) K-means where $k = b$, (3) Entropy: select top-$b$ points that minimize $\sum_c p(y = c|x) \log p(y = c|x)$ (where $c$ is the class with the highest probability), (4) Var Ratios: select top-$b$ points that minimize $p(y = c|x)$ (where $c$ is the class with the highest probability), (5) BAIT [13]: solving the same objective as our method, select $2b$ points and then delete $b$ points, both in greedy way. Random and K-means are non-deterministic, we run each test 20 times. The other methods are deterministic and the only randomness is related to $S_0$. We performed several runs varying $S_0$ randomly and there is no significant variability in the results, so for clarity we only present one representative run. We present the classification accuracy on $U$ in the upper row of Fig. 3. We can observe that our method consistently outperforms other methods across all experiments. K-means, one of the most popular methods due to each simplicity significantly underperformed FIRAL. It is worth noting that the random selection method serves as a strong baseline in the experiments of ImageNet-50, where our method initially outperforms Random but shows only a marginal improvement in later rounds. But random selection underperforms in CIFAR-10. In the lower row of Fig. 3, we show the images selected in the first round on ImageNet-50 for BAIT and FIRAL. Images selected in other methods

and other datasets can be found in Appendix G.2. One way to qualitatively compare the two methods is to check the diversity of the samples: in the 50-sample example BAIT samples only 21/50 classes; FIRAL samples 37/50 classes. This could explain the significant loss of performance of BAIT in the small sample size regime.

## 6 Conclusions

We presented FIRAL, a new algorithm designed for the pool-based active learning problem in the context of multinomial logistic regression. We provide the performance guarantee of our algorithm by deriving a excess risk bound for the unlabeled data. We validate the effectiveness of our analysis and algorithm using experiments on synthetic and real-world datasets. The algorithm scales linearly in the size of the pool and cubically on the dimensionality and number of classes—due to the eigenvalue solves. The experiments show clear benefits, especially in terms of robustness of performance across datasets, in the low-sample regime (a few examples per class).

One limitation of our algorithm is the reliance of a hyperparameter, $\eta$, derived from the learning rate in the FTRL algorithm. There are a large body of work in online optimization about the adaptive FTRL algorithm (e.g., [16]), which eliminates the need for such hyperparameter. In our future work, we will investigate the integration of adaptive FTRL and evaluate its impact on the overall performance of FIRAL. By exploring this avenue, we aim to enhance the flexibility and efficiency of our algorithm. Another parameter is the number of rounds to use in batch mode, but this we have just set to a small multiple of the number of classes. Other extensions include more complex classifiers and combination with semi-supervised learning techniques.

## 7 Acknowledgements

This material is based upon work supported by NSF award OAC 2204226; by the U.S. Department of Energy, Office of Science, Office of Advanced Scientific Computing Research, Applied Mathematics program, Mathematical Multifaceted Integrated Capability Centers (MMICCS) program, under award number DE-SC0023171; and by the U.S. National Institute on Aging under award number R21AG074276-01. Any opinions, findings, and conclusions or recommendations expressed herein are those of the authors and do not necessarily reflect the views of the DOE, NIH, and NSF. Computing time on the Texas Advanced Computing Centers Stampede system was provided by an allocation from TACC and the NSF.

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
