# OpenReview forum: "FIRAL: An Active Learning Algorithm for Multinomial Logistic Regression"
_NeurIPS.cc/2023/Conference — NeurIPS 2023 poster_

### Official Review · Reviewer_PwwV · 2023-06-11

**Soundness:** 3 good
**Presentation:** 3 good
**Contribution:** 3 good
**Rating:** 7
**Confidence:** 3

**Summary:**

The authors first prove that the excess risk of multinomial logistic regression with under subgaussian data distribution is lower and upper bounded by terms involving the ratio of the Fisher information of the unlabeled data and the Fisher information of the labeled data.
The authors then propose an algorithm for active learning, based in FIR minimization, and prove a performance guarantee for it.
Experimental results on simulated and real world datasets demonstrate well the performance of the proposed approach.

**Strengths:**

1. The proposed algorithm stands on a solid mathematical background.
2. The experimental results demonstrate the performance benefits over competitive methods.

**Weaknesses:**

1. The manuscript is very technical and not easy to read.
2. Comparison to competitive methods in terms of computation is missing.

**Questions:**

1. Can the authors refer to the capabilities of their algorithm in terms of scale?


**Limitations:**

Yes

---

> ### Author Rebuttal · Authors · 2023-08-10
>
> We thank reviewer PwwV for the review and comments!
> * **Not easy to read:**
> We acknowledge that it might be hard to comprehend Section 4.3 for readers unfamiliar with regret minimization. We have tried explain the overal merit of our approach while leaving most technical details in Appendix. We are open to any suggestions that could enhance the readability of our paper.
>
> * **Comparison in computation:**
> We interpret the question to refer the computational complexity of the algorithm. Our current implementation of FIRAL is much more expensive than random, k-means, and uncertainty-based methods. Its cost is comparable to BAIT. However, (1) we have not optimized some algorithmic details of FIRAL; and (2) FIRAL and BAIT significantly outperform the other methods.
>
> * **Scale capabilities:**
> Please see our general response.

---

> > ### Comment · Reviewer_PwwV · 2023-08-14
> > **Update to review**
> >
> > I have read the comments made by the other reviewers, as well as the author's rebuttal.
> >  One concern I have is regarding the high complexity of the algorithm in terms of $\tilde{d}$, which is prohibitive for most practical senses.
> > While I still recommend accepting the paper, I am waiting to read the post-rebuttal comments of the other reviewers and decide whether I should update the score from 7 to 6 in light of this issue.

---

### Official Review · Reviewer_B8jR · 2023-07-03

**Soundness:** 3 good
**Presentation:** 3 good
**Contribution:** 2 fair
**Rating:** 6
**Confidence:** 3

**Summary:**

This paper studies active learning when the underlying data distribution follows the multinomial logistic model. This paper considers the pool-based setting and designs an algorithm to select the sample to query in a batch fashion. There are two main contributions: (1) The paper shows that the excess risk is lower and upper bounded by the Fisher Information Ratio. (2) propose an algorithm to select the samples to optimize the Fisher Information Ratio and thus minimize the expected risk. Theoretical analysis and experiments are conducted to validate the proposed approach.

**Strengths:**

The strengths of the paper are as follows:
+ This paper shows the relationship between expected excess risk and the fisher information ratio for the multinomial logistic regression model
+ This paper proposes an algorithm to maximize the fisher information ratio
+ The technical part of this paper is well-written and easy to follow

**Weaknesses:**

The weaknesses of the paper are as follows:

- unclear comparison with existing work and novelty: my main concern about the paper is the novelty compared with existing literature.
   -  comparison with [11]: as mentioned by the authors, the non-asymptotic relationship between the FIR and excess risk has been provided in [11] for the generalized linear model. However, I am a little bit confused by the claim in lines 43-44 that "... the assumption that the loss function is strictly self-concordant (assumption 2 in [11]), which does not hold for the logistic regression case." Although the logistic regression loss is not strictly self-concordant, it is a general self-concordant function [27]. Based on the general self-concordance, it seems to me one can also bound the gap between the Hessians of two points as Assumption 2 in [11]. The authors also use the general self-concordant property (Proposition 31) to prove Theorem 3. I think it would be better if the authors could highlight that logistic loss is a general self-concordant function and highlight the novelty compared with [11].
   - comparison with [14]: it seems to me the method used to optimize the FIR is largely established on [14]. Proposition 7 is similar to a combination of Eq (8) and Proposition 3.1 in [14]. Both papers use the regret analysis of FTRL to show the lower bound in Proposition 8.  I think the paper would have a more significant influence if the author could highlight the main challenge of applying the analysis of [14] to the logistic regression model.

- about assumption in Theorem 3: Theorem 3 relies on the condition that $H_p(\theta_*)\leq H_q(\theta_*)$. It is unclear to me how large this constant would be and how to verify the condition in practice since $\theta_*$ is generally unknown.

- about the labeled dataset: the algorithm requires a labeled dataset $S_0$ to estimate the unknown parameter $\theta_*$. However, it is unclear to me how to obtain such a labeled dataset and when it is sufficient to obtain a good approximation of $\theta_*$ for the following active learning. As shown in Theorem 4, the number of labeled data will affect the coefficient $\alpha_0$ appearing in the exponential term that can be potentially large. I think it would be nice if the author could provide a more detailed discussion on the effect of the initial labeled data on the proposed algorithm.

- about the constants in Theorem 4: since $\alpha_0$ and $\alpha_n$ are related to $n_0$ and $n$, I am not sure they are appropriate to be considered constants in the proposed bound.







**Questions:**

- could you highlight the main technical challenges and contributions of the paper beyond the existing work? (please refer to the first point of weaknesses for more details)
- when the assumption on the Hessian $H_p$ and $H_q$ required by Theorem 3 is satisfied? How large is the constant $\sigma$?
- could you provide more discussion on the influence of the labeled dataset on the proposed method? (please refer to the third point of weaknesses for more details)

**Limitations:**

There are still some limitations in the paper:
- realizable assumption: the theoretical analysis is established on the realizable assumptions such that the data distribution follows the logistic regression model.
- tightness of the excess risk: the tightness of the proposed theoretical bound is not discussed.

---

> ### Author Rebuttal · Authors · 2023-08-10
>
> We thank reviewer B8jR for the review and comments!
> * **Novelty of Theorem 3 compared to [11]:**
> 	* Our primary contribution concerning Theorem 3 in comparison to the work presented in [11] is the generalization to sub-Gaussian distributions for the points. The proofs in [11] rely on an assumption of a bounded support for the points where we do not use this assumption.  We discussed this distinction in Lines 132--137 in our original submission. We remark that the bounded assumption is not explicitly stated in [11]. But it is required for the proofs in [11] to be valid. Specifically, item 5 of Assumption 1 in [11] assumed that  $\nabla L(y|x, \theta^*)$ is bounded by a constant $L_1$. For multinomial logistic regression, by the expression of the loss gradient in our paper (Equation (39)), we can infer that [11] assumes that the point $x$ is in a bounded domain.
> 	* Another contribution of our work on Theorem 3 in comparison to [11] is that we empirically demonstrate the excess risk bounds derived in Theorem 3 using synthetic datasets (as detailed in Section 5 and Appendix G.1).
> 	* Regarding the concern raised by the reviewer on Assumption 2 in [11] pertaining to multinomial logistic regression, we maintain our position that _Assumption 2 in [11] can not be deduced by the results in [27]_. Let us explain. Using the notation established in our paper, Assumption 2 of [11] asserts that there exits $L_4>0$ such that  $-L_4 \lVert\Delta\rVert_2 {\bf H}_p({\theta^\ast}) \preceq {\bf H}_p (\theta) - {\bf H}_p({\theta^\ast}) \preceq L_4 \lVert\Delta\rVert_2 {\bf H}_p({\theta^\ast})$, where $\Delta\triangleq \theta - \theta^\ast$.  We can interpret Equation (6) in Proposition 1 of [27] as implying $\big(e^{-R\lVert\Delta\rVert_2}-1\big){\bf H}_p({{\theta}^{\ast}}){\preceq}{{{\bf H}_p}(\theta)-{{\bf H}_p}({\theta^\ast})}{\preceq}\big(e^{R\lVert\Delta\rVert_2}-1\big){\bf H}_p({\theta^\ast})$, where $R>0$ corresponds to the constant delineated in Proposition 1 of [27].  We can get the lower bound of Assumption 2 in [11] from this relation. But we can not deduce the upper bound of Assumption 2 in [11] since $e^{R\lVert\Delta\rVert_2} - 1$ can not be upper bounded by $L_4 \lVert\Delta\rVert_2$  for constant $L_4$.
> * **Novelty of FIRAL compared to [14]:**
> In Lines 53 to 58 we discussed the two main challenges of applying the regret minimization approach in [14] for optimal design to our active learning setting for multinomial logistic regression. We appreciate the reviewer for raising this concern and agree on the need for a more detailed discussion on challenges and our contributions. Below are the main points:
>     - Algorithmically, there are technical details that are needed in extending the approach presented in [14] to formulate the point selection objective within our specific context. In [14], they work with a loss matrix that is essentially a rank-1 matrix $x x^\top$ for some $x$ in $d$-dimension, resulting in a  straightforward point selection objective (as seen in Line 9 of Algorithm 1 in [14]). In contrast, the loss matrix in our scenario, denoted as $\widetilde{{\bf H}}(x_{i_t})$  (Equation (15)),  possesses a minimum rank of $c-1$ and can even be a full-rank matrix, contingent upon the labeled points from prior rounds. In order to manage the complexity associated with directly utilizing objective (19), we delve into the structure of the Fisher information matrix and employ algebraic techniques to establish an equivalent objective (23) that is more computationally efficient. We believe that our analysis is more technical compared to the  situation addressed in [14].
>     - Theoretically, the task that presents the most technical challenge compared to [14] when establishing the performance guarantee for our algorithm is _Proposition 9_.  The corresponding bound derived is Lemma 3.2 in [14].  The distinction between the characteristics of the loss matrices  significantly complicates the derivation of such a general bound in Proposition 9.  We have to use several algebraic tricks to obtain the  bound for the near-optimal performance guarantee (see Appendix F.4). In our opinion, our proof is not trivial,  which is perhaps corroborated by comparing the length of our proof with the proof of Lemma 3.2 in [14].
>     - Empirically, we conduct active learning experiments on real-world datasets and demonstrate that our algorithm outperforms the compared baselines.
> * **Parameter $\sigma$:**
> We thank the reviewer for pointing this out. This is a typo in our manuscript, $\sigma>1$ should be $\sigma >0$ in Theorem 3 (also Theorem 32 for detailed version). We do not require $\sigma>1$ in our proof. We apologize for the confusion caused by this typo. We will fix it in the new version.
> * **Influence of initial labeled dataset:**
> We believe that the influence of the initial labeled dataset on the performance of active learning depends on the specific dataset in question. In our numerical experiments, using CIFAR-10 as an example, we randomly select one labeled point from each class to construct $S_0$. This approach ensures an equitable starting point for all methods, enabling a fair comparison. We agree that in practice, the selection of initial data points to label, absent any label information, can exert a significant impact on the initial performance of active learning, particularly when the budget for $|S_0|$ is severely restricted, say $1c$ or $2c$. There are several methods for selecting $S_0$: Random sampling, K-means clustering, or optimal experimental design [14]. In Table 1 of the rebuttal PDF, we compare the performance of these different sampling methods for $S_0$ for  $|S_0|=20$ and its effect to FIRAL. In Table 1, we observe that the effect of $S_0$ on the performance of FIRAL  is not significant (for this particular experiment).
> * **Constants in Theorem 4:**
> Thanks for pointing this out. We will modify our expression regarding to these two constants in a more proper way.

---

> > ### Comment · Reviewer_B8jR · 2023-08-14
> >
> > Thank you for your detailed reply. The rebuttal has largely addressed my concerns regarding the novelty of this paper, particularly in its comparison with [14]. While there are similarities in the high-level ideas between this paper and [14] when addressing the optimization problem (13), I agree with the authors that the derivation of (23) and the analysis to reach Proposition 9 need specific technical adjustments tailored to the multinomial logistic model. Given this, I would like to adjust my score to 6.
> >
> > Turning to the comparison with prior work [11], I retain some reservations about the statement concerning the self-concordant property. I understand that the generalized self-concordant property (Proposition 31) used in this paper is different from Assumption 2 in [11], particularly in its coefficient ($e^{R\Vert\Delta\Vert_2} - 1$ here, as opposed to $L_4\Vert \Delta\Vert$ in [11]). Yet, both conditions essentially utilize the self-concordant function to shift the Hessian matrix, as seen in the formulation $H_p(\theta) - H_p(\theta^*)\leq f(\Delta) H_p(\theta^*)$. It seems to me the difference in the coefficients $f(\Delta)$ just leads to different coefficients in the final bound. As evidence, the coefficient $f(\Delta) = (e^{\Vert \Delta\Vert}-\Vert \Delta\Vert-1)/\Vert \Delta\Vert^2$ finally leads to the $(e^\alpha -\alpha -1 )/\alpha^2$ term in Theorem. In this sense, I think the two conditions still share many similarities. I suggest the authors provide a more clear comparison with the previous work by just simply saying that "the strictly self-concordant does not hold for the logistic regression case."

---

> > > ### Author Response · Authors · 2023-08-16
> > >
> > > Thank you for considering our response and increasing the score!
> > >
> > > We appreciate your new suggestions regarding the comparison to the previous work [11], specifically in terms of comparing our Theorem 3 with Lemma 1 of [11]. In the revision of our draft we will: (1) remove the statement about the self-concordance property of [11], (2) remark the comparison with [11] by summarizing key technical aspects at the end of section 3 (i.e. lines 132-137) in our original submission, and (3) provide a more detailed comparison in the appendix.
> > >
> > > But let us give some more details on the differences between our approach and [11].
> > >
> > > First, we realized that the self-concordance property (Assumption 2 in [11]) was _not_ used in deriving Lemma 1 of [11]. (Even though it is included as a premise in the Lemma 1 of [11], it remains unused in the proof.) Thus, we will remove the statement about the self-concordance property of [11] in our introduction section.  Lemma 1 in [11] derives the excess risk bounds by using the Taylor theorem on $L_p(\theta_n) - L_p(\theta_\ast)$. In contrast, our approach to Theorem 3 involves initially establishing the general self-concordance of $L_p(\theta)$ and subsequently applying Proposition 31.
> > >
> > > Second, there exists a difference in the manner by which the spectral approximation relations among multiple Hessian matrices are derived in Lemma 1 of [11] and our Theorem 3. This is due to the different assumptions used in our work and [11]. In Lemma 1 in [11], the  spectral approximation relations are derived by using Bernstein-type inequalities that depend on the regularity conditions listed in Assumption 1 of [11].  In our work, the more general sub-Gaussian assumption (Assumption 1 in our submission) incorporated within our Theorem 3 introduces some technical challenges to our derivation. For example, in order to obtain the spectral approximation relations presented in Equations (111) and (136), we  initially establish high probability bounds as detailed in Proposition 33 (and subsequently Corollary 34). Within this proposition, we employ a covering argument to infer spectral properties for random matrices associated with the property (3) of our Lemma 2.
> > >
> > > We cannot thank you enough for taking so much time to review our work and giving valuable suggestions to our paper!

---

### Official Review · Reviewer_zZMa · 2023-07-06

**Soundness:** 3 good
**Presentation:** 3 good
**Contribution:** 3 good
**Rating:** 6
**Confidence:** 3

**Summary:**

This paper proposes a novel active learning algorithm called FIRAL for pool-based active learning in multinomial logistic regression. The paper investigates the theory and algorithms for pool-based active learning and compares FIRAL to other active learning methods in terms of classification error. The authors use finite sample analysis to establish FIRAL-based bounds for the excess risk in the case of multinomial logistic regression with sub-Gaussian assumption for the point distributions. The experiments conducted on various datasets show that FIRAL outperforms other active learning algorithms in terms of classification error.

**Strengths:**

The equation and proof make sense. Good introduction on this part.

Convinced theory.

**Weaknesses:**

Not enough introduction on related work.

The target task is logistic regression classifier. I think it is simple and has been researched for many years. If this method can be applied on more complex classifiers or segmentation, it would be better.

No ablation study on hyperparameter, such as \eta.

**Questions:**

In Line 67, the logistic regression classifier here is very simple, is it possible to apply this method on more complex classifiers, such as ResNet, Transformer, which need more data to train?

What is the meaning of \theta_*? The optimal parameters to predict y?

Could you compare with uncertainty-based active learning method, such as "Towards better uncertainty sampling: Active learning
with multiple views for deep convolutional neural network"?

Could you do experiments on the whole ImageNet data set, not just ImageNet-50?

**Limitations:**

The notation is not constant. For example, the subscripts of \theta_0 and \theta_n have different meaning.

Some writing errors. Line 17 "we use the sample the b points". Line 20 "choose q in order minimize"... Please check them.

---

> ### Author Rebuttal · Authors · 2023-08-10
>
> We thank reviewer zZMa for the review and comments!
>
> * **Introduction on related work:**
> We will add more discussion on related work in the new version.
>
> * **Classifier choice:**
> We used the multinomial logistic regression classifier in order to be able to conduct the theoretical analysis. We believe our non-asymptotic analysis in Theorem 3 and our algorithm FIRAL can be extended to other Generalized Linear Models with the change of Fisher information matrix. We remark that we did use deep-learning based contrastive learning for feature extraction, on which we then apply our classifier. Comparison of our current methodologies  with a deep learning classifier that directly uses FIRAL would be interesting but also challenging, as it would be very hard to train a deep learning network with just a handful of examples. This comparison is important but beyond the scope of the paper.
>
> * **Selection Hyperparameter $\eta$:**
> Please see our general response to all reviews.
>
> * **Question regarding the definition of $\theta_*$**
>  In original submission we introduced $\theta_*$ in lines 71--72:  we assume that the true distribution  $p(y|x)$ is given by the multinomial logistic regression with $\theta_*$.
>
> * **Compare with uncertainty-based active learning methods**
> We did compare uncertainty-based active learning methods such as "entropy" and "Var Ratios" in our paper (Figure 3 in the original submission). "Var Ratios"  is also referred as "Confidence" in some literature.
>
> * **Whole ImageNet dataset**
> Unfortunately, the $O(b \tilde{d}^3)$ scaling of FIRAL's current implementation  makes it very expensive to apply the scheme to the entire ImageNet as it would require solving thousands of 40K-by-40K eigenvalue problems (if we use 40 features for each data point). However, we would like to emphasize that this is not a fundamental limitation of FIRAL. As we discussed in our common response, FIRAL can be significantly sped up.

---

> > ### Comment · Reviewer_zZMa · 2023-08-21
> >
> > Thank you for your rebuttal. All my questions have been solved. I have increase the rating. Please write better in the final paper.

---

### Official Review · Reviewer_C5TF · 2023-07-09

**Soundness:** 4 excellent
**Presentation:** 3 good
**Contribution:** 3 good
**Rating:** 6
**Confidence:** 3

**Summary:**

This paper develops a pool-based active learning method for multinomial logistic regression, following a long line of work using the Fisher Information Ratio as a criterion for active set selection. The paper establishes FIR to tightly (within constants) characterize excess risk (Theorem 3), so that this can be used to select a set of points for active learning. This method is guaranteed pretty well (Theorem 4).

**Strengths:**

It's nice that the evaluations include strong baselines like BAIT; this is a valuable contribution because of the computational efficiency of the proposed method. \tilde{d} is relatively small (hundreds) in useful regimes, where the proposed method would be quite efficient. I would not be entirely convinced that it performs better than BAIT without the empirical evaluations - but since they are present and look promising, this seems like a useful method.

The setting of multinomial logistic regression is at once specific enough to be useful here, and general enough to be interesting (could be any discrete label). So the extent to which theory meets practice here is uncommon and laudable - fantastic problem selection, and approach (with the FIR being the "right" quantity for other exponential families).

**Weaknesses:**

The sparsification problem in (14) in Section 4.3 is where a lot of looseness in the theory creeps in; optimizing the eigenvalue for sparsification is a roundabout way of selecting points. I am a little puzzled by the relaxation of Section 4.3; the natural relaxation of the convex program (13) is a randomized rounding procedure optimizing over z \in [0,1]^{m} , rather than optimizing over the positive orthant as in (14). I am also curious about the choice of FTRL for sparsification. It works well enough, but initially was quite confusing in the exposition.

In Figure 2, dashed constant-slope lines would help readability a lot.

The main theoretical results are stated in an unnecessarily complex way, I believe.
- The complex dependence on \alpha is not really necessary to fully write out in the theorem statements. Fig. 1 handles this well, but in order to interpret it in context, you should stress that n needs to be O(\tilde{d}) for \alpha to be low and the bounds to be tight. The complex interplay between these variables could overall use better explanation.
- Theorem 4's use of the \lesssim makes it difficult to interpret the (1+\epsilon) prefactor on the right-hand side, and I had to rummage in the appendix and proofs to understand what the result is trying to express.

**Questions:**

- The parameter settings for the real-world experiments are unclear right now (e.g. for \eta); the included code has \eta = 200, but is this ok for all datasets?
- I'd like to see the results with multinomial logistic regression on pretrained frozen embeddings, i.e. "linear probing." The theory here might actually reflect practice closely enough to make the results noteworthy. This type of approach would strengthen the paper a lot, and opportunities to do so with theory in non-toy settings are not so common.

**Limitations:**

Yes

---

> ### Author Rebuttal · Authors · 2023-08-10
>
> We thank reviewer C5TF for the review and comments!
> * **Relaxation problem:**
> The reviewer is correct. The constraint on Equation (14) should be $z\in[0,1]^m$. We are sorry for the confusion caused by the expression. We will fix it in the new version.
>
> * **Why use FTRL for sparsification problem?:**
> We use FTRL because it allows to prove optimality of our algorithm.  Specifically, with FTRL we can establish a lower bound in Equation (18) for $\lambda_{\min}\big(\sum_{t\in [b]}\widetilde{\bf H}(x_{i_t})\big)$. Our goal in point selection consequently becomes the maximization of this lower bound, encapsulated by Equation (19), which is equivalently expressed in Equation (23). For the reason why Equation (18) is satisfied, we provided an explanation in Appendix F.3. The primary steps can be outlined as follows: (1) FTRL algorithm provides an upper bound (Equation (250)) for the  cumulative regret (defined in Equation (245)). (2) After applying Lemma 43, we can get the lower bound (Equation (254)) for the minimum eigenvalue of the summed loss matrices, i.e. ${\lambda_{\min}}\big(\sum_{t \in [b]} {\bf F_t}\big)$. (3) Since this bound is satisfied for any symmetric semi-positive definite loss matrix with dimension $\tilde{d}$, we can let loss matrix ${\bf F_t}=\widetilde{\bf H}({x_{i_t}})$, thereby arriving at the obtained lower bound in Equation (18).
>
> * **Clarifying main theoretical results:**
> Thanks for pointing them out. We will address them promptly by revising the content and discussion of Theorems 3 and 4 to ensure it becomes more concise and appropriately expressed.
>
>  * **Selecting $\eta$:**
>   Please see our general response.
>
> * **Experiment on pretrained frozen embeddings:**
> We compared various active learning methods on CIFAR-10 using frozen pretrained embeddings with a dimension of 512. The results of this comparison are presented in Figure 3, in the rebuttal PDF.   In the plot, it is evident that FIRAL outperforms other methods  methods until 200 points are labeled. After this point, uncertainty-based methods, BAIT and FIRAL demonstrate comparable performance levels.

---

### Official Review · Reviewer_3UFt · 2023-07-31

**Soundness:** 4 excellent
**Presentation:** 3 good
**Contribution:** 3 good
**Rating:** 7
**Confidence:** 3

**Summary:**

The authors present theory and algorithms for training multinomial logistic regression models in the pool-based active learning setting; how we should choose $b$ extra points to label from a pool of unlabeled ones, so that when we train a model including the newly acquired labeled points the excess risk of the classifier is minimized. The provide with novel asymptotic lower and upper bounds for the excess risk under sub-gaussian assumptions on the point distribution, and they show that excess risk is in $\Theta$ of the Fisher Information Ratio between the proposal and point distributions. This justifies algorithms which select points based on the minimization of Fisher Information Ratio. They further devise an algorithm for $(1 + \epsilon)$-optimal minimization of the Fisher Information Ratio objective. The algorithm depends on approximating the oracle predictive model $p(y|x, \theta^*)$ by a pretrained classifier, a continuous convex relaxation of the NP-hard discrete problem of point selection, and a greedy regret minimization procedure for sparsifying the continuous solutions, while maintaining bounds for the optimality of the final solution. Finally, they perform extensive experimentation on synthetic and image classification benchmarks to demonstrate that 1) their bounds are numerically valid, 2) their method outperforms chosen baselines.

**Strengths:**

1.  They provide asymptotic lower and upper bounds for the excess risk **under sub-gaussian assumptions** on the point distribution, and they show that excess risk is in $\Theta$ of the Fisher Information Ratio between the proposal and point distributions. This weakens bounded domain assumptions found for similar bounds in the literature.
2.  The employed sparsification methodology in the provided algorithm seems to be novel in the context of pool-based active learning, while maintaining the ability to have performance guarantees.
3.  Numerical experiments are extensive and demonstrate that the algorithm outperforms the compared baselines. Even in the larger scale ImageNet-50 case, with pretrained feature extractors, their method provides with some benefits over random sampling.

**Weaknesses:**

1.   Portion of the effort and presentation goes into reducing the requirement of performing eigendecompositions of a $\tilde{d} \times \tilde{d}$ matrix for each unlabeled point to performing eigendecompositions of $(c - 1) \times (c - 1)$. However, there is still need to perform $b$ eigendecompositions of  $\tilde{d} \times \tilde{d}$ matrices in order to use the FTRL algorithm. As a result, complexity is still $O(\tilde{d}^3)$.
2.   It seems that in more demanding cases (like ImageNet-50 presented in paper), there are diminishing returns from using an active learning setting like FIRAL over randomly sampling from unlabeled samples. This may discourage practitioners from implementing a pool-based active learning approach over random sampling from unlabeled samples, in order to avoid computational complexity issues and implementation overheads. This seems to be significant as the authors chose to further embed features extracted from pretrained networks with SimCLR to an even lower dimensional space using spectral embeddings. Evaluating the proposed algorithm under more practical considerations, it would be interesting to see the baseline of random sampling (and even K-means) from the unlabeled pool using instead the full $D$ dimensional extracted features from the pretrained network (in the cases of CIFAR10 and ImageNet-50); baselines which are certainly computationally possible to execute.

**Questions:**

### Questions
*  On a more general note, active learning might prove itself more advantageous when unlabeled points U come from a different distribution than the actual distribution $p$ that we want to evaluate the excess risk under. Such example is the setting of training with class-imbalanced sets (or more generally datasets which have been collected under biased processes), which is a common issue with uncurated large-scale datasets collected in-the-wild. Changing the setting slightly, e.g. by assuming access to a small set S of curated class-balanced samples, might incur benchmarks that are even more favorable to using approaches like FIRAL. What are the authors thoughts about such baselines and problems?

### Typos and Edits
* Line 17: “we use **the** sample the b points” -> “we use to sample these b points”
* Line 25: “is bounded above and below **by** FIR”. Also, there should be a citation about FIR at that place.
* Line 36: “sub-Guassian”
* Line 38: NP-hard problem needs a citation there
* Line 80: $V_p$ definition has a $q$, which should be $p$

**Limitations:**

See Weaknesses and Questions.

---

> ### Author Rebuttal · Authors · 2023-08-10
>
> We thank reviewer 3UFt for the review and comments!
> *  **On the algorithm complexity:**
> Please see our general response.
>
>
> *  **Experiment on pretrained frozen embeddings:**
> We compared various active learning methods on CIFAR-10 using frozen pretrained embeddings with a dimension of 512. The results of this comparison are presented in Figure 3, which is included in the rebuttal PDF. In the plot, it is evident that FIRAL outperforms the  methods up to the 200 labeled points budget. After this point, uncertainty-based methods, BAIT and FIRAL demonstrate comparable performance levels.
>
>
> *  **Experiment on imbalanced dataset:**
> Thank you for pointing this to us, this is an excellent point. Indeed,  active learning helps a lot with imbalanced datasets.  We tested FIRAL on a class-imbalanced CIFAR-10 (Figure 2 in the rebuttal PDF). We observe that FIRAL and BAIT are not sensitive to class imbalance in the active pool and  significantly outperform the other methods.

---

> > ### Comment · Reviewer_3UFt · 2023-08-11
> > **Convincing rebuttal, thank you for extra experiments**
> >
> >  -  **On the algorithm complexity**: This is maintained as a weakness of the current paper, but I am convinced that with sufficient effort in terms of selection of solvers we can keep this kinda tractable in the large-scale scenario. I am looking forward for future work applying this in large-scale.
> >  -  **Experiment on pretrained frozen embeddings** and **experiment on imbalanced dataset**, thank you for providing these convincing experiments.
> >
> > I am raising the score to 7.

---

### Author Rebuttal · Authors · 2023-08-10

# General Response:
We thank the reviewers for their careful read of our paper, their comments, and  their suggestions. We have fixed all the typos mentioned in the reviews. We have submitted responses to individual researchers. We also submitted a PDF with additional results.

There were two issues that were raised by two reviewers: the complexity of the algorithm and the selection of the hyperparameter $\eta$. We address them below.

###  **Complexity of algorithm**
In our original submission, we stated the computational complexity of FIRAL at line 231.  Recall that $c$ is the number of classes, $d$ is the number of features per point, $\tilde{d}=d(c-1)$, $m$ is the number of unlabeled points in the pool, and $b$ is the number of points to select for labeling.  We use a dense-matrix SPD eigenvalue solver,  and  direct linear solver. These yield the complexity of FIRAL, as implemented now: $O(m c^3 + b \tilde{d}^3 + m \log m \tilde{d}^3).$  The $c^3$ term is due to Eq 23 (trace); the $b$-term is the SPD-eigenvalue solve in Alg 1, line 8 for  $\left(\sum_{s=1}^t \tilde{H}_s\right)$.  The $\log m$ comes from the $m$ linear solves (Alg 2, line 5, linear solve for $\Sigma$ and the trace) and $\log m$ expected iterations ($T$ in Alg 2, line 2).

The performance of these steps can be significantly sped-up by (1) using matrix-free iterative solvers and (2) exploiting the structure of $H(x)$ (Eq 10, line 164). The storage of $H(x)$ is just $d+c$ if we only store ${\bf h}(x)$ and $x$. A matrix-free vector multiplication with $H(x)$ can be done in $\tilde{d}$ time using the structure of Eq 10. This structure can be exploited to accelerate computations with $\tilde{H}$ and $\Sigma^{-1}$. The trace calculations, eigenvalue solves, and linear solves can be done approximately using Krylov methods and randomized algorithms.

 For example, Eq 23 and Alg 2 trace calculations can be approximated using randomized trace estimation that requires approximate matrix-vector multiplications; and can be readily parallelized on multi-GPU architectures. The linear solve in Alg 2 can be done using the Conjugate Gradient method. The eigenvalues solve can be done using an iterative Lanczos solver. In this an approximation of $\nu_t$ could be in principle be computed in $O(b \tilde{d})$ cost.

 We believe that by introducing fast algorithms and a more scalable high-performance implementation, the algorithm could scale to million of points and thousands of classes.  Our goal in this paper was simply to establish the theoretical correctness and optimality of FIRAL;  and to demonstrate its feasibility using modest-scale experiments. Scaling FIRAL to large datasets requires significant effort and merits a separate analysis. Such effort is beyond the scope of this paper.

### **Selection of hyperparameter $\eta$:**
The selection of $\eta$ is done use a grid search and doesn't require labeling. (In our code this search is manual, but it can be automated.)  In our original submission, we discussed tuning $\eta$ in Appendix G.2 (Lines 1132 to 1134): we tested different values of $\eta$ to determine the one that maximizes $\lambda_{\min}\big(\sum_{t\in[b]}\widetilde{ \bf H}(x_{i_t})\big)$, as this is the main goal of our sparsification problem.

In the rebuttal PDF in Figure 4,  we report results of an ablation study on $\eta$ and included the results in Figure 4 of the newly submitted file: we plotted $\lambda_{\min}\big(\sum_{t\in[b]}\widetilde{ \bf H}(x_{i_t})\big)$ along with the accuracy under different values of $\eta$ in the first two rounds of the active learning tests on CIFAR-10. In our experiments, we explored different values of $\eta$ to determine the one that maximizes $\lambda_{\min}\big(\sum_{t\in[b]}\widetilde{ \bf H}(x_{i_t})\big)$, as this is the main goal of our sparsification problem (parameter tuning in Appendix G.2).  It is evident that $\lambda_{\min}\big(\sum_{t\in[b]}\widetilde{ \bf H}(x_{i_t})\big)$ serves as a valuable guide in selecting an appropriate $\eta$ that leads to good accuracy performance.

---

### Decision · Program_Chairs · 2023-09-21

**Decision:**

Accept (poster)

**Comment:**

This paper introduces FIRAL, an algorithm with theoretical guarantees under weaker assumptions than prior work and convincing experimental results. Given the unanimous recommendation for acceptance from the four reviewers, I recommend acceptance.